# Analysis of Anti-Cancer and Anti-Inflammatory Properties of 25 High-THC Cannabis Extracts

**DOI:** 10.3390/molecules27186057

**Published:** 2022-09-16

**Authors:** Dongping Li, Yaroslav Ilnytskyy, Esmaeel Ghasemi Gojani, Olga Kovalchuk, Igor Kovalchuk

**Affiliations:** Department of Biological Sciences, University of Lethbridge, Lethbridge, AB T1K 3M4, Canada

**Keywords:** *Cannabis sativa*, flower extracts, delta-9-tetrahydrocannabinol, anti-cancer property, anti-inflammatory property

## Abstract

*Cannabis sativa* is one of the oldest cultivated plants. Many of the medicinal properties of cannabis are known, although very few cannabis-based formulations became prescribed drugs. Previous research demonstrated that cannabis varieties are very different in their medicinal properties, likely due to the entourage effect—the synergistic or antagonistic effect of various cannabinoids and terpenes. In this work, we analyzed 25 cannabis extracts containing high levels of delta-9-tetrahydrocannabinol (THC). We used HCC1806 squamous cell carcinoma and demonstrated various degrees of efficiency of the tested extracts, from 66% to 92% of growth inhibition of cancer cells. Inflammation was tested by induction of inflammation with TNF-α/IFN-γ in WI38 human lung fibroblasts. The efficiency of the extracts was tested by analyzing the expression of COX2 and IL6; while some extracts aggravated inflammation by increasing the expression of COX2/IL6 by 2-fold, other extracts decreased inflammation, reducing expression of cytokines by over 5-fold. We next analyzed the level of THC, CBD, CBG and CBN and twenty major terpenes and performed clustering and association analysis between the chemical composition of the extracts and their efficiency in inhibiting cancer growth and curbing inflammation. A positive correlation was found between the presence of terpinene (pval = 0.002) and anti-cancer property; eucalyptol came second, with pval of 0.094. p-cymene and β-myrcene positively correlated with the inhibition of IL6 expression, while camphor correlated negatively. No significant correlation was found for COX2. We then performed a correlation analysis between cannabinoids and terpenes and found a positive correlation for the following pairs: α-pinene vs. CBD, p-cymene vs. CBGA, terpenolene vs. CBGA and isopulegol vs. CBGA. Our work, thus, showed that most of high-THC extracts demonstrate anti-cancer activity, while only certain selected extracts showed anti-inflammatory activity. Presence of certain terpenes, such as terpinene, eucalyptol, cymene, myrcene and camphor, appear to have modulating effects on the activity of cannabinoids.

## 1. Introduction

*Cannabis sativa* is a plant with a long history of consumption as food and medicine. Most of the medicinal properties, however, are either anecdotal or are reported at the pre-clinical level.

Extracts from cannabis flowers contain many ingredients, including major cannabinoids such as delta-9-tetrahydrocannabinol (THC), CBD, cannabigerol (CBG) and cannabinol (CBN), as well as many minor cannabinoids, terpenes and terpenoids, flavonoids, phenols, fatty acids and many more [1]. Not all of these ingredients have medicinal properties and the extracts vary hugely in the composition of these ingredients; thus, cannabis is not generic when used as medicine. However, several ingredients, including primarily THC and CBD, are considered to be active [2], since there is significant pre-clinical and clinical evidence about their activity.

Delta-9-tetrahydrocannabinol (THC) is one of the main cannabinoids in cannabis; it has many properties, including anti-cancer, anti-inflammatory, analgetic and others [3,4]. However, not all cannabis extracts that are high in THC appear to be equally effective [5]. Previously, our lab and others reported drastic differences in the anti-inflammatory properties of various cannabis extracts and demonstrated modulating effects of various minor cannabinoids and terpenes [6,7]. Specifically, we recently found that the combination of several terpenes downregulated pro-inflammatory cytokines with efficiency similar or greater that single cannabinoids [6]. Analysis of six different high-THC extracts showed different efficiencies against inflammation, with four being effective, one being neutral and one increasing the expression of pro-inflammatory cytokines [7].

It is hypothesized that other ingredients in the extracts have modulating effects on THC. Such ingredients may be CBD itself, or other minor cannabinoids or molecules, such as terpenes. Indeed, terpenes were shown to regulate the activity of major cannabinoids in vitro and in vivo [8,9].

Here, we have profiled 25 different high-THC extracts for anti-cancer and anti-inflammatory properties. We also profiled cannabinoids and terpene concentrations and performed correlation analysis with anti-cancer and anti-inflammatory activities, and identified positive and negative modulating effects.

## 2. Results

### 2.1. Analysis of Cannabinoids Content in 25 Cannabis Varieties

Analysis of cannabinoid content in flowers showed that all twenty-five varieties were high-THC varieties, with a total THC level between 7.1 and 15.1% (Figure 1A). The level of total CBD varied from 0.13% to 1.3%, and CBGA from 0.18% to 1.3% (Figure 1A).

We next extracted the flowers using ethyl acetate and measured the cannabinoid content. The concentration of THC ranged from 26.5 to 36.9%, CBD from 0.58 to 3.62% and CBGA from 0.64 to 1.71% (Figure 1B). We noted that extraction had different efficiency in the tested cultivars; the extract enrichment with THC varied from 1.95-fold in cultivar #17 to 4.59 in cultivar #11 (Table 1). A similar trend was observed for CBD and CBGA, but overall, the enrichment was much higher for CBD as compared to THC and CBGA; the average enrichment in CBD was 4.81-fold, while in THC and CBGA, it was 2.94 and 2.97, respectively (Appendix A). This experiment demonstrated that flowers from different cultivars are different in their capacity to release cannabinoids into the solvent, and that CBD is extracted more efficiently with our extraction method.

### 2.2. Analysis of Terpene Content in 25 Cannabis Varieties

Analysis of terpenes showed that, on average, δ-limonene was the most abundant terpene, followed by β-caryophyllene (Appendix A). Ocimene and eucalyptol were less abundant among the detected terpenes. Correlation analysis showed a very weak positive correlation between the total level of cannabinoids and total level of terpenes (r = 0.14).

### 2.3. Inhibitory Effect of Extracts on Cancer Cells

Anticancer activity was measured using HCC1806 breast squamous cell carcinoma treated with all 25 extracts. We first analyzed the range of concentrations and identified 0.007–0.03 μg/μL as a range of active concentrations that can inhibit the cell growth (data are not shown). We selected three concentrations, 0.007 μg/μL , 0.01 μg/μL and 0.015 μg/μL , for the analysis of all twenty-five extracts. We found that all extracts could inhibit the growth of HCC1806 cells in a dose-dependent manner; extract #24 was the strongest with 92% inhibition, while #16 was the weakest, with 66% inhibition at 0.015 μg/μL after 96 h of treatment (Figure 2; Appendix A). To test whether these concentrations had any effect on normal cells, we treated BJ-5ta normal human foreskin cells. It should be noted, however, that BJ-5ta is a hTERT-immortalized fibroblast cell line, and therefore, as a transformed cell line may not be a representative of how the primary cell line would respond. We found that extracts also inhibit the growth of normal cells, but to a lesser degree—10.1% in response to extract #24, and 37.5% in response to extract #16. 

Since we found inhibitory effects of the extracts on both, cancer and normal cells at 0.015 μg/μL , we wanted to test whether the lower concentration would spare normal cells. We also extended the analysis to the 120 h check point.

The analysis of the effect of extracts #21–25 showed that at 120 h, there was a further reduction in the growth of cancer cells (Table 1); in response to the extract #24, cancer cells were inhibited by 96.5%, while normal cells only by 6.85%. Thus, it appears that extract #24 is one of the best, as it inhibits the cancer growth to a much higher extent than it does normal cells.

We also noted that at 120 h, 0.01 μg/μL was not toxic to normal cells, while still inhibiting the growth of cancer cells. In fact, 0.007 and 0.01 μg/μL slightly improved the growth of normal cells. 

We first hypothesized that anti-cancer properties are dependent on the presence of THC, as a major component of the extracts (~31.4% of the dry weight of the extract and ~90% of the total cannabinoid content). To our surprise, the correlation analysis showed no correlation between the anti-cancer effect and THC or total cannabinoid content, r = 0.017 and r = 0.057, respectively, while a weak positive correlation was found for CBGA (r = 0.26). As the average THC content was 31.4%, the amount of THC in 0.015 μg/μL of the extract was 0.00471 μg/μL . To confirm the role of THC in anti-cancer effects, we exposed HCC1806 cells to 0.00157 μg/μL (5 μM), 0.00314 μg/μL (10 μM) and 0.00471 μg/μL (15 μM) of THC alone. We found a dose-dependent reduction in the growth of cancer cells, with up to 45% inhibition observed at the highest concentration equivalent to the amount of THC in the extracts (Figure 3). Since, on average, extracts inhibit cancer growth by ~80%, and since we found no correlation between the THC levels in the extract, it appears that THC alone is not responsible for the inhibition of the growth of cancer cells.

### 2.4. Anti-Inflammatory Properties of 25 Cannabis Extracts

To test the anti-inflammatory properties of the extracts, we used the WI38 lung epithelial cells or HSIEC intestinal epithelial cells. First, we induced inflammation in WI38 cells by treating them with TNF-α/IFN-γ for 24 h and measured the induction of inflammation by Western blot analysis of COX2 and IL6 pro-inflammatory cytokines (Figure 4A–C). The analysis showed that while some extracts decreased inflammation (decreasing the expression of COX2 and IL6 after the induction with TNF-α/IFN-γ), many extracts caused the opposite effect (Table 2). On average, there was no effect on COX2 (1.02-fold reduction) and a small effect on IL6 (1.32-fold reduction) in WI38 cells (Table 2). 

We then selected the five best performing extracts and compared their effect on two different cell types, WI38 and HSIEC. We found that anti-inflammatory properties of the selected extracts were much stronger on WI38 cells than on HSIEC cells; the effect on COX2 in WI38 was especially pronounced, as on average, a 3.5-fold reduction in the expression of COX2 was observed (Figure 4 and Table 2).

These experiments demonstrated that only some high-THC extracts are efficient against inflammation (as measured by COX2 and IL6 levels), and that the extracts differ in their effect on the inflammation in lung and intestinal epithelial cells.

Similar to the effect on cancer cells, we decided to establish the contribution of the THC levels on the observed anti-inflammatory effect of the extracts. We induced inflammation in WI38 cells and treated them with THC at various concentrations, 0.3 ng/uL (0.0003 μg/μL or 1 μM), 0.00157 μg/μL (5 μM), 0.00314 μg/μL (10 μM), 0.00471 μg/μL (15 μM) and 0.00628 μg/μL (20 μM). The concentration of 0.00471 μg/μL represented the average concentration of THC in the extracts. We found that concentrations of up to 0.00314 μg/μL (10 μM) decreased the inflammation (more prominently in IL6), while higher concentrations increased it, and quite substantially (Figure 5). Therefore, the anti-inflammatory effect observed at 0.015 μg/μL (0.00471 μg/μL of THC) of extracts also cannot be explained by the presence of THC alone.

### 2.5. Correlation Analysis

Next, we attempted to identify the active ingredient(s) in the extracts that were responsible for the observed effects. First, we performed association analyses between total cannabinoids and single cannabinoids with anti-cancer or anti-inflammatory activities. No associations were found.

To select the best clustering model to use for our data, we calculated the cophenetic correlation coefficient (CCC)—a correlation between the distance matrix and cophenetic distance, by comparing three methods, ward.D2, complete and McQuitty. We found that the McQuitty method was the best, with CCC of scaled and filtered data of 0.86 (Appendix A). Another method used the Dunn index, which also showed that the McQuitty clustering method was the best (Appendix A). Clustering analysis revealed several major clustering groups, from single cultivars to as many as 10 cultivars in the main cluster (Figure 6A). It is worth noting that the most efficient extract, #24, clustered separately from the other cultivars.

Principal component analysis (PCA) of terpene composition showed that most cultivars are substantially different, with only cultivars #5 and #13 being relatively similar (Figure 6B; Appendix A).

We then applied the Gaussian generalized linear model to detect associations between terpenes and the anti-cancer activity of extracts. We found no association between total terpene levels and anti-cancer or anti-inflammatory activities. Analysis of single terpenes showed low positive (0.87) significant association between the presence of terpinene and anti-cancer activity (Figure 6C; Appendix A). It should be noted, however, that this association was influenced by a single extract with a very large amount of gamma-terpinene. Associations for eucalyptol and fenchone were much higher, 9.56 and 4.17, respectively, but they were not significant, pval_BH = 0.09 and 0.7, respectively. 

Similar association analysis between terpene composition and anti-inflammatory properties showed significant positive association between p-cymene concentration and reduction in IL6 expression (Appendix A); the same was observed for β-myrcene. Negative association was observed for camphor. Unfortunately, multiple comparison adjustment made these associations non-significant. For COX2, none of the associations were significant (Appendix A).

We then performed association analysis between the presence of individual terpenes and cannabinoids. Significant associations were found for the following pairs: α-pinene vs CBD, p-cymene vs CBGA, terpenolene vs CBGA and isopulegol vs CBGA (Figure 6D–G; Appendix A); linalool also scored high in association with CBGA (Appendix A).

## 3. Methods

### 3.1. Source of Flowers and Extract Preparation

*Cannabis sativa* flowers were received from Cannabis West Development Corp. CannaWest and Cloudburst Cannabis and were stored in the licensed facility at the University of Lethbridge (license number LIC-62AHHG0R77-2019). The exact names of cultivars are provided in Appendix A. According to CannWest, the plants were grown in comparable conditions, including vegetative and flowering stages; no additional information was available. Flowers were given arbitrary names from #1 to #25. Flowers from each cultivar were pulverized and three grams of the material was used for extraction. Powdered material was placed in a 250 mL Erlenmeyer flask and 100 mL of ethyl acetate was added. The flasks were incubated overnight in the dark at 21 °C with continuous shaking at 120 rpm. Extracts were then filtered, concentrated using a rotary vacuum evaporator and transferred to a tared 3-dram vial. To eliminate the residual solvent, the vials were kept in an oven overnight at 50 °C. Working extract stocks were prepared from the crude extracts, by dissolving 3–6 mg of crude extract in DMSO (dimethyl sulfoxide anhydrous, ThermoFisher Scientific, Toronto, Canada) to obtain the final concentration of 60 mg/mL; extracts were then stored at −20 °C. For the actual experiments with cancer cells or normal cells, appropriate cell culture media (RPMI + 10% FBS or EMEM + 10% FBS) were used to dilute the 60 mg/mL stock to make working medium. Extracts were sterilized using a 0.22 µm filter.

### 3.2. Analysis of Cannabinoids

Agilent Technologies 1200 Series HPLC system, equipped with a G1315C DAD, G1316B column compartment, G1367D autosampler, and G1312B binary pump was used to analyze the acidic and neutral forms of phytocannabinoids. The separation was performed on a Phenomenex Kinetex EVO C18 column (5 µm, 100 × 2.1 mm id) with a Phenomenex SecurityGuard ULTRA guard column. Instrument control, data acquisition, and integration were carried out with ChemStation LC 3D Rev B.04.02 software (Agilent Technologies, Santa Clara, United States). A 2 µL injection volume was used for all calibration standards (THC, CBD, THC-A, CBD-A, CBG, CBG-A, all Sigma-Aldrich, Montreal Canada) and sample analysis. The compound peaks were detected at 230 nm and 280 nm. Mobile phases consisted of 50 mM ammonium formate (pH 5.2) (Sigma-Aldrich) in HPLC grade water (ThermoFisher Scientific, Toronto, Canada) on the A side and 100% methanol (Fisher Chemical) on the B side, with a flow rate 0.3 mL/min. Two samples per each flower sample were analyzed, with two technical repeats per each sample.

### 3.3. Analysis of Terpenes

Terpene analysis was performed on dry flowers of cultivars #1–#25, using an 8610C GC coupled with a flame ionization detector (FID) from SRI Instruments at Canvas Labs (Vancouver, BC, Canada). 

### 3.4. Cell Culture

Human normal foreskin fibroblasts BJ-5ta (purchased from American Type Culture Collection, ATCC, Manassas, VA, USA) were cultured in Dulbecco’s Modified Eagle’s Medium, supplemented with 10% fetal bovine serum (FBS). BJ-5ta is a hTERT-immortalized fibroblast cell line. Human breast squamous cell carcinoma cells HCC1806 (purchased from Cell Biologics, Chicago, IL, USA) were cultured in the Epithelial Cell Medium/w Kit. Human primary small intestinal epithelial cells (HSIEC), purchased from Cell Biologics, were cultured in the Epithelial Cell Medium/w Kit. Human lung fibroblasts (WI-38), purchased from ATCC, were cultured in Eagle’s Minimum Essential Medium, supplemented with 10% FBS. All cell lines were incubated at 37 °C in a humidified atmosphere of 5% CO_2_. Mycoplasma contamination was regularly monitored using a Mycoplasma PCR Detection kit (Applied Biological Materials Inc., Richmond, BC, Canada) and eradicated using BM-Cyclin (Sigma-Aldrich, Darmstadt, Germany), according to the manufacturer’s instructions.

### 3.5. Screening of Anti-Inflammatory Cannabis Extracts

WI-38 and HSIEC cells grown to 80% confluency were treated with 10 ng/mL TNF-α/IFN-γ (Sigma), alone or in combination with 5 µM CBD or 0.01 µg/µL cannabis extracts or 1–20 µM THC, while 1% DMSO served as a control. At 48 h after treatment, cells were washed three times with cold PBS and whole cellular lysates were prepared and stored at −20 °C.

### 3.6. MTT Assay

Once the cells reached 70–90% confluency, 3 × 10^3^ BJ-5ta cells/well or 3 × 10^3^ HCC1806 cells/well were replated in 96-well plates. At 24 h after incubation, cells were treated with either 7.5 µg/mL or 15 µg/mL of extracts or with 5–15 μM THC; 1% DMSO served as a control. Cells were harvested every 24 h for five consecutive days. Assays were performed with 3-(4,5-Dimethylthiazol-2-yl)-2,5-diphenyltetrazolium bromide (MTT) using the Cell Proliferation Kit I (Roche Diagnostics GmbH) in triplicate, as described by the manufacturer. The spectrophotometric absorbance of samples was measured at 595 nm using a microtiter plate reader (FLUOstar Omega, BMG LABTECH, Offenburg, Germany).

### 3.7. Western Blot Analysis

The indicated cells were rinsed twice with ice-cold PBS and scraped off the plate in a RIPA buffer; 30–100 μg of protein per sample was electrophoresed on 8%, 10%, or 12% SDS-PAGE and electrophoretically transferred to a PVDF membrane (Amersham Hybond™-P, GE Healthcare, Arlington Heights, IL, USA) at 4 °C for 1.5 h. Blots were incubated for 1 h with 5% non-fat dry milk to block nonspecific binding sites, and subsequently incubated at 4 °C overnight, with 1:200 to 1:1000 dilution of polyclonal/monoclonal antibodies against COX2 (Cat# ab15191) or IL-6 (Cat# sc-130326) (all from Santa Cruz Biotechnology, Dallas, TX, USA). Immunoreactivity was detected using a peroxidase-conjugated antibody and visualized by an ECL Plus Western Blotting Detection System (GE Healthcare, IL, USA). The blots were stripped before re-probing with an antibody against GAPDH (Abcam, Cambridge, UK). Densitometry of the bands was measured and normalized with that of GAPDH using ImageJ.

### 3.8. Statistical Analysis

Two-tailed Student’s *t*-test was used to determine the statistical significance of the difference in the expression of COX2 and IL-6, as well as cell growth in the MTT assay. *p* < 0.05 was considered significant.

### 3.9. Correlation Analysis

The statistical analysis was conducted using R language version 4.0.1. The terpenes with concentrations close to 0 were removed from the analysis. The associations between scaled terpenes and cancer scores were detected using a generalized linear model (GLM) implemented as the *glm()* function implemented in R, with the family option set to *gaussian*. The resulting *p*-values underwent multiple test adjustments using the Benjamini-Hochberg method (cite: Benjamini and Hochberg, 1995, J. R. Statist. Soc.). 

### 3.10. Clustering Analysis

Filtered and scaled terpene concentrations were clustered using the *hclust()* R function with methods set to ward.D, complete or McQuitty. The distance measure in each case was *euclidean*. The best clustering method was selected based on the cophenetic correlation coefficent (CCC) and Dunn’s index. The clustering results were visualized as dendrograms, with heatmaps generated using the *pheatmap* R package. 

### 3.11. PCA Analysis

Principal components analysis (PCA) was conducted in R using the *prcomp()* function. PCA results were visualized as principal component plots with the *ggplot2* graphical R package.

## 4. Discussion

Here, we demonstrated that high THC cannabis extracts have strong effects on the growth of breast cancer cells but exhibit selective anti-inflammatory effects that were dependent on the extract used and pro-inflammatory marker analyzed. Our work, thus, showed that most of the high-THC extracts demonstrate anti-cancer activity, while only certain selected extracts showed anti-inflammatory activity. Correlation analysis showed that the presence of terpinene, and likely eucalyptol, positively correlated with anti-cancer activity, while the presence of p-cymene and β-myrcene positively correlated with IL-6 expression, and camphor negatively correlated with IL-6. No correlation between the concentration of any tested cannabinoid and anti-cancer or anti-inflammatory activity was found.

It is interesting to observe that THC demonstrated a stronger cell growth inhibitory effect as compared to the anti-inflammatory effect. We observed no correlation between the level of THC and anti-cancer or anti-inflammatory activity, suggesting other ingredients had strong modulating effects. It should be noted that the THC concentration was ~10-fold higher than the other cannabinoids combined, and, at the tested concentrations, would likely have a predominant effect on cancer cells. The concentration of CBD in all tested extracts was relatively low, and there was no correlation between the amount of CBD and anti-cancer activity or anti-inflammatory activity. However, extracts #3 and #4 that had the highest CBD content had above average anti-inflammatory activity. 

We also found a weak correlation between the concentration of CBGA and anti-cancer activity (r = 0.26), while no correlation was observed for anti-inflammatory properties. CBG was found to be effective in inhibiting proliferation of mouse melanoma and oral epithelioid carcinoma cells; in the latter, CBG was the most effective cannabinoid [10]. A recent paper tested the effect of CBG alone or in combination with other cannabinoids on multiple glioblastoma cell cultures; it was found that CBG and THC were comparable in their effect, but significantly subpar to the effect of CBD. Furthermore, combining CBG with CBD, but not THC, had a strong modulating effect on proliferation inhibition [11]. This suggests that CBG and CBD may be the modulatory cannabinoids affecting the anti-cancer activity of THC. Indeed, it was shown that a cocktail of cannabinoids present in a specific extract was much more efficient than THC alone, albeit, the crude extract that contained other molecules, such as terpenes, was more efficient in inhibiting the growth of glioblastoma cells [12]. Comprehensive analysis of response of more than 10 cancer lines, including brain and breast cancer, revealed that THC alone was inferior compared to extracts containing THC in inhibiting the cells growth in 15 out of 17 tests conducted [12,13,14,15]. 

A recent comprehensive review of research papers comparing the effect of various cannabinoids on various cancers showed that in most of the cases, CBD was more effective (had lower IC50 values) than THC in inhibiting cancer cell growth [13]. In addition, pure CBD was typically more effective than CBD-rich extracts [16,17]. A similar trend was observed in breast cancer cell xenografts in mice; CBD was more effective alone [16], while THC was less effective [15] as compared to extracts with CBD or THC.

Cannabinoids are not the only molecules found in cannabis extracts that exhibit anti-cancer properties. Other components, including terpenes and flavonoids, were shown to inhibit cancer growth in vitro and in vivo. A number of terpenes, including myrcene, caryophyllene, pinene, humulene, limonene and many more have demonstrated to have anti-cancer properties and cytotoxic effects, inducing apoptosis, cell cycle arrest, decreasing cell migration and invasion (reviewed in [18]).

In our work, we found that terpinene, eucalyptol and fenchone also likely have the anti-breast cancer effect acting alone or in combination with THC. Eucalyptol demonstrated to have anti-tumor effects in vitro and in vivo [19]; it induced cell cycle arrest, apoptosis and changes the expression of number of genes involved in cancer progression [20]. The anti-proliferative effect of terpinene on breast cancer cells was associated with the induction of apoptosis [21]. Fenchone derived from the essential oil extracted from *Mesosphaerum sidifolium* also demonstrated anti-tumor activity in previous studies [22].

However, how the terpenes potentiate the anti-cancer effects of cannabinoids remains unclear. Several studies demonstrated that it is unlikely that such effects occur through the activation of CB1 and CB2 [23]. In addition, it was shown that α-pinene, β-pinene, β-caryophyllene, linalool, limonene, β-myrcene or α-humulene do not modulate the activity of cannabinoids through TRPA1 and TRPV1 channels either [24]. So, it is likely that terpenes modulate the activity of cannabinoids through the interaction with other receptors or through nonreceptor-mediated mechanisms. One of the most recent studies reported that the cannabis extracts were more potent than single cannabinoids on the inhibition of the growth of breast cancer cells, and that the combination of the five most abundant terpenes (β-caryophyllene, α-humulene, nerolidol, linalool and β-pinene) with cannabinoids did not reconstitute the effect of the whole extract [15]. Thus, it is quite possible that other minor terpenes or cannaflavins have modulatory effects on cannabinoids.

### 4.1. Anti-Inflammatory Effect—THC vs. Other Cannabinoids

As mentioned above, the effect of high-THC extracts on pro-inflammatory cytokines was not as significant as its antiproliferative effects on breast cancer cells, with some extracts having stronger effects than others, suggesting that other cannabinoids, terpenes or unidentified molecules have strong potentiating effects.

Previous publications demonstrated that THC and its synthetic analogues had anti-inflammatory activity in different cells, tissues and in human in vivo [25,26,27]. Comparison of THC to other cannabinoids showed that CBD and THC were similarly effective in the inhibition of NLRP3 inflammasome activation, following LPS treatment of human THP-1 macrophages and primary human bronchial epithelial cells [28].

Previous work showed that CBD was somewhat better than THC in suppressing the expression of ACE2 and TMPRSS2—two proteins required for infection with SARS-CoV2 [6]. Furthermore, it was shown that several of the most abundant terpenes were able to downregulate these proteins, and, in combination with THC and CBD, were able to reconstitute the activity of the whole extract [6]. THC had a stronger effect on the reduction in COX2, IL-6 and IL-8 expression as compared to CBD, while the extracts and combination of cannabinoids with the top five terpenes were more efficient than corresponding amounts of THC or CBD when tested on lung fibroblast WI-38 cells [6].

While it may not be clear what components of extracts are more effective in the reduction in inflammation, it is important to test many of them; in several cases, we tested the efficiency of extracts with comparable levels of CBD and THC and found them to vary greatly in their ability to inhibit ACE2 and TMPRSS2 and to reduce key pro-inflammatory markers, such as COX2, IL-6 and IL-8 [5,7]. Correlation analysis carried out for seven extracts showed a weak positive correlation between THC (r = 0.24) and anti-inflammatory effects and weak negative correlation for CBD (r = −0.29) in the human 3D skin artificial EpiDermFT tissue model [5]. One way to evaluate the efficacy of various extracts in curbing inflammation is to use gene expression analysis algorithms, such as those published recently [29]. It allows us to evaluate overall capacity to reduce inflammation, based on the changes in the transcriptome of ~120 inflammation-related genes.

Our correlation analysis showed that terpinene, eucalyptol, p-cymene, β-myrcene and camphor positively correlated with the anti-inflammatory effect of extracts. These terpenes, and others, including trans-caryophyllene [30], α-bisabolol [31], β-caryophyllene [32], geraniol [33], and valencene [34], have been shown to attenuate the production and the release of cytokines, such as IL-1β, IL-6, and TNF-α. As mentioned above, it is not known how terpenes affect inflammation, as they do not appear to interact with most of the cannabinoid receptors tested [23,24]. However, as in the case of cancer, it appears that terpenes do potentiate the effect of cannabinoids on various cytokines.

It should be noted that often, single cannabinoids are more potent than the combination of cannabinoids or cannabinoids with terpenes [35]. Previous work has shown that CBD and CBG alone reduced LPS-induced inflammation in guinea pigs, while a combination of the two compounds (1:1) did not [35]. Comparison of the effect of CBD standard and high CBD cannabis extract demonstrated superior activity of pure CBD on several parameters of inflammation, including IL-6 and IL-8 expression in lung cells [36]. It is likely that such a discrepancy may be due to the difference in the model used, treatment type, concentration of molecules, and type of cytokines analyzed.

### 4.2. Correlation between Cannabinoids and Terpenes

Analysis of the correlation between the level of cannabinoids and terpenes led to intriguing outcomes. The only correlation for CBD was found between α-pinene and CBD; no correlation was found for THC. As for CBGA, a positive correlation was found with p-cymene, linalool, γ-terpinene and terpinolene, while a negative correlation was found for isopulegol and β-myrcene.

Recently, the analysis of 108 cultivars revealed that β-caryophyllene, β-myrcene, α-pinene, limonene and terpinolene were the most abundant and that cannabis varieties (typically high in THC) differed from hemp varieties (low in THC, and relatively high in CBD) only by a higher concentration of those terpenes, rather than by the presence of unique terpenes [37]. In another work, THC-A levels positively correlated with γ-selinene, β-selinene, α-gurgujene, γ-elemene, selina-3,7(11)-diene, and β-curcumene, while CBD-A levels negatively correlated with these terpenes in high THC varieties. In contrast, in high CBD varieties, CBD-A positively correlated with the levels of β-eudesmol, γ-eudesmol, guaiol, α-bisabolene, α-bisabolol, and eucalyptol, while THC-A negatively correlated with these terpenes [38]. To observe the opposite correlation of CBD and THC for so many terpenes is somewhat surprising and may be artificial; indeed, the authors only analyzed three cultivars per each chemotype (CBD or THC). In contrast, Hillig reported that the β-eudesmol, γ-eudesmol, and guaiol were enriched in high THC varieties, although no correlation analysis was carried out [39].

In one previous study, hierarchical clustering and PCA performed on 21 cannabis varieties showed that high CBD varieties also typically had a high level of α-pinene [40]. We could not find any information for CBGA; hence, we are likely to be the first to report the correlation between the presence of CBGA and several terpenes.

## 5. Conclusions

In sum, while THC may be effective in the inhibition of the growth of breast cancer cells, other cannabinoids, such as CBD or CBG, as well as terpenes have strong modulating effects.

It appears that even 25 extracts are not sufficient to clearly delineate the active ingredient(s) in cannabis or modulating effects of the other minor components. This suggests the complexity of all the ingredients, perhaps indicating that many of them are active and have modulation capacities. In addition, it may suggest that, perhaps, other untested ingredients, such as cannflavins, have a strong effect on cancer and inflammation [41]. Our work is not without the following limitations: in the future, we have to test other minor cannabinoids alone or in combination; we have to also identify flavonoids and test their activity; and, we have to test other cancer cell lines as the effects are likely to be cancer-specific [18]. Furthermore, we used ethyl acetate for the extraction, and it is possible that other solvents would allow us to change the profile of the extracts, and thus may have different medicinal effects. One other limitation of our study is the use of transformed normal cells—BJ-5ta cells are an hTERT-immortalized fibroblast cell line. These cells may respond to extracts differently from the primary cells.

## Figures and Tables

**Figure 1 molecules-27-06057-f001:**
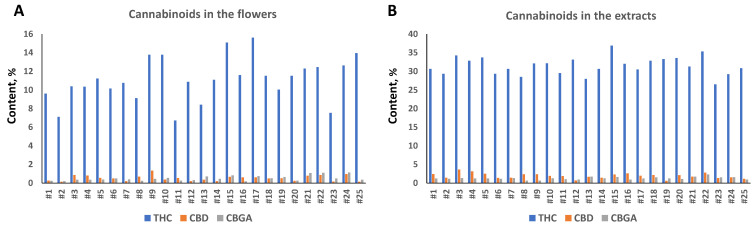
Total THC, CBD and CBG content in the flowers (**A**) and extracts (**B**). Concentration is shown in the percentage of total weight of dry flowers or extracts prepared from flowers of 25 varieties.

**Figure 2 molecules-27-06057-f002:**
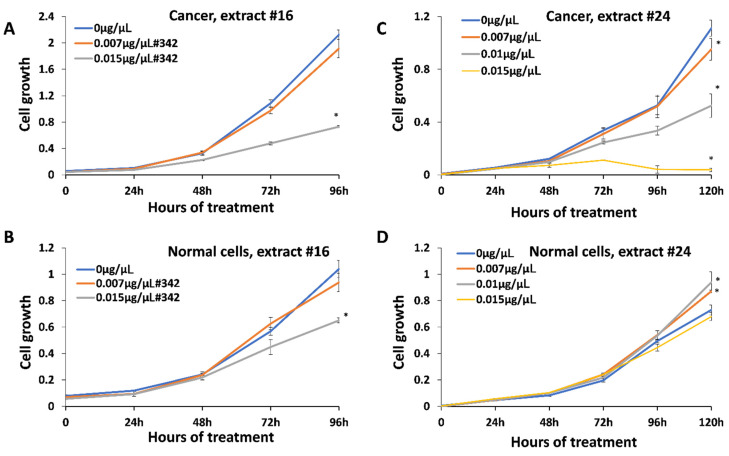
Inhibition of growth of breast cancer cell line HCC1806 and normal cell line BJ-5ta. (**A**)—growth curve of HCC1806 cells in response to extract #16; (**B**)—growth curve of BJ-5ta cells in response to extract #16; (**C**)—growth curve of HCC1806 cells in response to extract #24; (**D**)—growth curve of BJ-5ta cells in response to extract #24. Cells were treated with 0, 0.007 and 0.015 μg/μL of the above-mentioned extracts for various periods of time, from 0 to 120 h. MTT assay was performed and data were expressed as an average (from 3 independent replicates) with SD. Asterisks indicate significant (*p* < 0.05) difference from control (0 μg/μL ).

**Figure 3 molecules-27-06057-f003:**
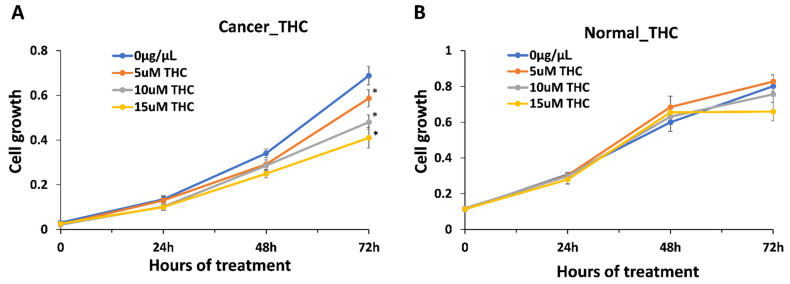
Inhibition of growth of breast cancer cell line HCC1806 and normal cell line BJ-5ta in response to THC. (**A**)—growth curve of HCC1806 cells in response to 5, 10 and 15 μM of THC; (**B**)—growth curve of BJ-5ta cells in response to 5, 10 and 15 μM of THC. Y axis shows arbitrary units of cell growth, while X axis shows the time of treatment. MTT assay was performed and data were expressed as an average (from 3 independent replicates) with SD. Asterisks indicate significant (*p* < 0.05) difference from control (0 μg/μL ).

**Figure 4 molecules-27-06057-f004:**
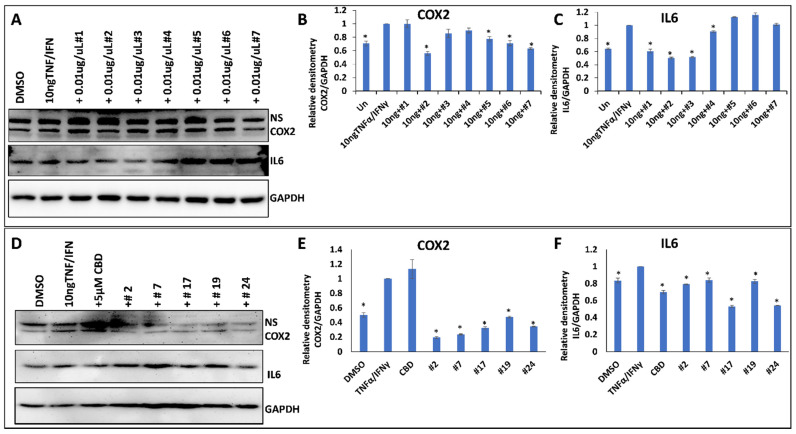
The effect of selected extracts on the expression of COX2 and IL6 in WI38 cells treated by 10 ng/mL TNF-α/IFN-γ. (**A**)—Western blot image of COX2, IL6 and GAPDH in response to DMSO, 10 ng/mL TNF-α/IFN-γ, or TNF/IFN together with one of the extracts, from #1 to #7. (**B**)—ImageJ calculated the densitometry of COX2 for samples #1–#7. (**C**)—ImageJ calculated the densitometry of IL-6 for samples #1–#7. (**D**)—Western blot image of COX2, IL6 and GAPDH in response to DMSO, 10 ng/mL TNF-α/IFN-γ, 10 ng/mL TNF-α/IFN-γ plus 5 µM CBD, or TNF/IFN together with one of the extracts—#2, #7, #17, #19 and #24. (**E**)—ImageJ calculated densitometry of COX2 for samples #2, #7, #17, #19 and #24. (**F**)—ImageJ calculated densitometry of IL-6 for samples #2, #7, #17, #19 and #24. Data are shown as average relative to GAPDH, calculated from 3 independent measurements, with SE. Asterisks show significant difference from TNF-α/IFN-γ treatment (*p* < 0.05). NS—non-specific binding.

**Figure 5 molecules-27-06057-f005:**
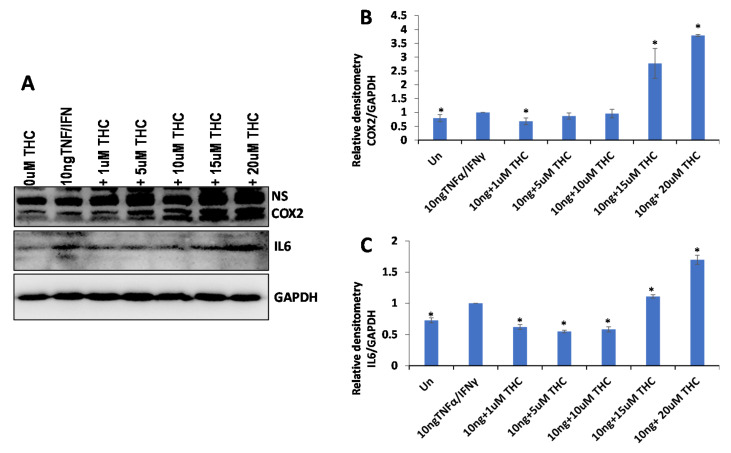
The effect of various concentrations of THC on the expression of COX2 and IL6 in WI38 cells treated with 10 ng/mL TNF-α/IFN-γ. (**A**)—Western blot image of COX2, IL6 and GAPDH in response to DMSO, 10 ng/mL TNF-α/IFN-γ, or TNF-α/IFN-γ together with different concentrations of THC. ImageJ calculated the densitometry of COX2 (**B**) and IL6 (**C**) relative to GAPDH, calculated from 3 independent measurements, shown as averages with SE. Asterisks show significant difference from TNF-α/IFN-γ treatment (*p* < 0.05).

**Figure 6 molecules-27-06057-f006:**
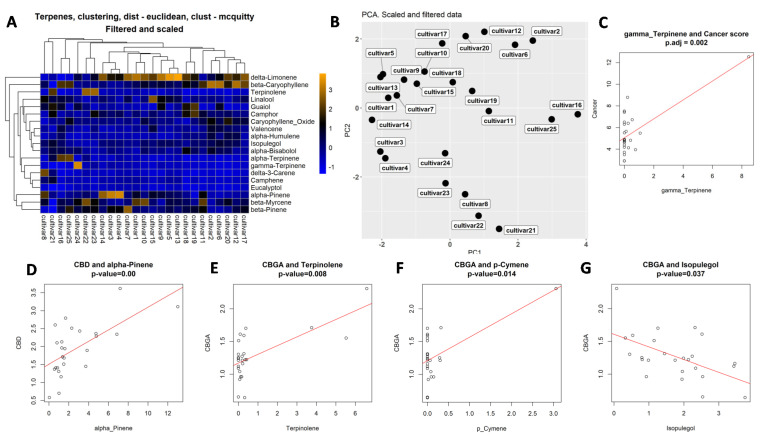
Clustering and correlation analysis. (**A**)—clustering of cultivars according to their terpene concentrations. (**B**)—PCA analysis according to the terpene concentration. Correlation analysis indicating correlation between the concentration of gamma-terpinene and anti-cancer activity (**C**), between CBD and alpha-pinene levels (**D**), CBGA and terpinolene (**E**), CBGA and p-cymene (**F**) and CBGA and isopulegol (**G**). Open circles indicate individual extracts.

**Table 1 molecules-27-06057-t001:** Inhibition of cancer and normal cells by cannabis extracts at 120 h post treatment.

	Cancer, %	Normal, %	Cancer, %	Normal, %
#1	76.08	47.40		
#2	79.21	12.00		
#3	85.12	39.10		
#4	79.38	52.20		
#5	86.50	53.90		
#6	77.17	36.40		
#7	81.85	31.90		
#8	84.59	40.80		
#9	81.72	40.80		
#10	84.13	37.30		
#11	79.59	43.50		
#12	78.77	41.40		
#13	78.31	33.80		
#14	71.35	35.20		
#15	82.27	36.90		
#16	65.52	37.50		
#17	65.75	20.10		
#18	79.25	48.70		
#19	75.73	16.80		
#20	73.05	40.00		
#21	88.61	21.80	95.86	24.50
#22	84.35	18.00	95.23	32.70
#23	74.68	−11.00	90.28	5.90
#24	92.03	10.10	96.50	6.85
#25	86.70	16.20	96.44	20.05
Average	79.67	32.03	94.86	18.00

Numbers show average (from 3 plates, with 3 measurements per plate) percent inhibition of cell growth as estimated by MTT assay. Two sets of data are presented, first on all twenty-five extracts and second on extracts #21–#25. Negative number indicates growth promotion.

**Table 2 molecules-27-06057-t002:** Anti-inflammatory effect of cannabis extracts.

Cultivar Name	WI38		WI38		HSIEC	
	COX2	IL6	COX2	IL6	COX2	IL6
#1	1.00	1.65				
#2	1.79	1.99	5.16	1.26	0.50	1.61
#3	1.17	1.93				
#4	1.11	1.10				
#5	1.29	0.89				
#6	1.40	0.86				
#7	1.58	0.99	4.24	1.19	0.48	1.74
#8	0.89	2.12				
#9	1.10	1.65				
#10	0.71	1.87				
#11	0.48	1.53				
#12	0.33	1.53				
#13	0.26	1.96				
#14	0.80	0.76				
#15	0.81	0.59				
#16	0.65	0.71				
#17	1.39	1.24	3.09	1.89	1.71	1.84
#18	0.76	0.56				
#19	1.00	0.33	2.11	1.21	1.78	1.53
#20	0.83	1.16				
#21	0.93	0.47				
#22	1.26	3.43				
#23	1.05	0.91				
#24	1.22	1.78	2.90	1.85	0.43	1.30
#25	1.63	1.07				
Average	1.02	1.32	3.50	1.48	0.98	1.61

Numbers show fold difference in the level of COX2 or IL6 proteins in the samples treated with TNF-α/IFN-γ versus control. For WI38, two sets of data are shown, one using all twenty-five extracts, and one using extracts #2, #7, #17, #19 and #24.

## Data Availability

All data are available in the main text or the Appendix A.

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
