# Peer review of "Analysis of Anti-Cancer and Anti-Inflammatory Properties of 25 High-THC Cannabis Extracts"

_molecules, 2022, doi:10.3390/molecules27186057_

Round 1

Reviewer 1 Report

In the reviewed manuscript, the authors discuss the anti-cancer and anti-inflammatory properties of cannabis extracts. It is a very interesting topic, especially in the context of discussions on the use of medical marijuana in other diseases. Currently, there are many indications for the use of canobinoids also in inflammatory diseases. The article is interesting, the research methodology is at a high level and includes tests such as MTT, Western Blot, HPLC, which are now widely used and considered sufficient to determine the biological properties of natural and synthetic compounds. The discussion and conclusions that the anti-cancer and anti-inflammatory properties do not depend on one substance are interesting, but are the resultant of the actions of individual components of the extract. The manuscript lacked information on the differences between the individual samples of the plant material. There is no analysis as to whether the differences in the effect of the extracts result from, for example, where the plants were grown. Also, line 79 has fiver instead of five, and line 479 has TCH instead of THC.

Author Response

In the reviewed manuscript, the authors discuss the anti-cancer and anti-inflammatory properties of cannabis extracts. It is a very interesting topic, especially in the context of discussions on the use of medical marijuana in other diseases. Currently, there are many indications for the use of canobinoids also in inflammatory diseases. The article is interesting, the research methodology is at a high level and includes tests such as MTT, Western Blot, HPLC, which are now widely used and considered sufficient to determine the biological properties of natural and synthetic compounds. The discussion and conclusions that the anti-cancer and anti-inflammatory properties do not depend on one substance are interesting, but are the resultant of the actions of individual components of the extract. The manuscript lacked information on the differences between the individual samples of the plant material. There is no analysis as to whether the differences in the effect of the extracts result from, for example, where the plants were grown. Also, line 79 has fiver instead of five, and line 479 has TCH instead of THC.

Our response:

Thank you for your kind words. We have added the information on plants. Flowers were provided by Cannabis West Development Corp. – the company stated that growth conditions were comparable but did not disclose any further information to us.

We also fixed typos.

Reviewer 2 Report

The authors Li et al. in their MS entitled “Analysis of anti-cancer and anti-inflammatory properties of 25 high-THC cannabis extracts” summarize analysis of 25 cannabis extracts containing high level of delta-9-tetrahydrocannabinol (THC) to inhibit proliferation of selected cell carcinoma lines and inflammation response in human fibroblast cell lines. Primarily the authors analyzed the level of selected cannabinoids and twenty major terpenes. Then they performed clustering and association analysis between the chemical composition of extracts and their efficiency in inhibiting cancer growth and inflammatory response. Authors conclude that most of high-THC extracts demonstrate anti-cancer activity, while only several extracts showed anti-inflammatory activity. They suggested that the presence of certain terpenes such as terpinene, eucalyptol, cymene, myrcene and camphor modulated activity of cannabinoids. The topic of this research is highly interesting. However, the study is purely observational, without attempt to find a mechanism how the extracts affect the cell proliferation or response to inflammatory mediators. Further, the experimental set up and overall design of the in vitro tests using selected tumor cell lines and normal fibroblasts brings major limitations. Following issues were found to limit the study. The responses to the raised questions should be elaborated into the text body to improve clarity and the meaning of the study.

Major:

- Why was ethyl acetate employed to prepare the extract from the flowers? With other organic solvents a different composition of the extracts would be obtained.

- How the selected concentration for testing is relevant to dose that the cell in vivo can be exposed to?

- The comparison between selected “cancer” cell line and selected “normal” cell line lacks deeper meaning. How the cancer cell line differs from normal cell line, which mechanisms of cell proliferation are regulated differently in cancer cell line compared to normal one? Why authors selected these cancer cell lines? The “normal cell line” proliferate under these in vitro conditions that mean it is not “normal”. Mitogens in fetal serum deregulate control of cell proliferation and to proliferate similarly to cancer cells. Thus, it is not clear what information can be obtained from the comparison of “normal” and “cancer” cells. Authors have to focus on mechanisms that are deregulated differently.

- It is not clear why the authors choose to stimulate cells by combination of pro-inflammatory mediators TNF-alpha and interferon gama. Are these cytokines activating a signaling pathways that are affected by cannabinoids or present terpenes?

Minor

- The short version of the abstract is missing the info about deep chemical analysis of 25 different varieties.

- the row 25 the symbol gama is missing

- the row 79 – a typo – fiver

- Data in table 1 are missing values for deviation. Data from similar experiments should be averaged and expressed with values showing deviation of the data (SD or SEM).

- Titles for the figures “cancer#xx” or “normal#xx” are in my opinion unclear suggesting that 25 different cancer cell lines were tested. Authors should find other type of labels for graphs.

- A clarification of labels for bars and western blot samples in figures 4 and 5 is needed - It is not clear which samples were treated by which combinations.

- Supplementary data – title “Supplementary Table 2. Concentration of terpenes in flowers” should be “Supplementary Table 2. Concentration of terpenes in in extract from flowers”

 - Legends of tables explaining the associations (the supplementary tables 5- 8) should briefly explain what index of association of expressed and from which data.

 - Specification of flowers manufacturers should be included.

- In most cases the symbol gama for interferon gama is missing.

- The use of abbreviations should be unified, and the list of abbreviations should be provided.

Author Response

Reviewer #2

The authors Li et al. in their MS entitled “Analysis of anti-cancer and anti-inflammatory properties of 25 high-THC cannabis extracts” summarize analysis of 25 cannabis extracts containing high level of delta-9-tetrahydrocannabinol (THC) to inhibit proliferation of selected cell carcinoma lines and inflammation response in human fibroblast cell lines. Primarily the authors analyzed the level of selected cannabinoids and twenty major terpenes. Then they performed clustering and association analysis between the chemical composition of extracts and their efficiency in inhibiting cancer growth and inflammatory response. Authors conclude that most of high-THC extracts demonstrate anti-cancer activity, while only several extracts showed anti-inflammatory activity. They suggested that the presence of certain terpenes such as terpinene, eucalyptol, cymene, myrcene and camphor modulated activity of cannabinoids. The topic of this research is highly interesting. However, the study is purely observational, without attempt to find a mechanism how the extracts affect the cell proliferation or response to inflammatory mediators. Further, the experimental set up and overall design of the in vitro tests using selected tumor cell lines and normal fibroblasts brings major limitations. Following issues were found to limit the study. The responses to the raised questions should be elaborated into the text body to improve clarity and the meaning of the study.

Our response:

Thank you for your comments. We will address them below.

Major:

- Why was ethyl acetate employed to prepare the extract from the flowers? With other organic solvents a different composition of the extracts would be obtained.

Our response:

We completely agree with you. Different solvents would likely extract different molecules and that might affect the results. Ethyl acetate is also an organic solvent, commonly used for extraction of active ingredients from medicinal plants (Lazarjani, M.P., Young, O., Kebede, L. et al. Processing and extraction methods of medicinal cannabis: a narrative review. J Cannabis Res 3, 32 (2021)). The main message of the manuscript is that molecules other than cannabinoids affect the results. Use of a different organic solvent is unlikely to dramatically influence the cannabinoids content in the extract, but would affect the presence of other molecules. We agree, it would be interesting in the future to test other solvents. We now discuss this in the text.

- How the selected concentration for testing is relevant to dose that the cell in vivo can be exposed to?

Our response:

We actually do not know. I do not believe that cannabis was ever used for treatment of human cancers. That is why it is important to first identify what the effect is in vitro and identify active components. The concentrations that we used were those that were effective in our in vitro studies.

- The comparison between selected “cancer” cell line and selected “normal” cell line lacks deeper meaning. How the cancer cell line differs from normal cell line, which mechanisms of cell proliferation are regulated differently in cancer cell line compared to normal one? Why authors selected these cancer cell lines? The “normal cell line” proliferate under these in vitro conditions that mean it is not “normal”. Mitogens in fetal serum deregulate control of cell proliferation and to proliferate similarly to cancer cells. Thus, it is not clear what information can be obtained from the comparison of “normal” and “cancer” cells. Authors have to focus on mechanisms that are deregulated differently.

Our response:

You are raising very interesting discussion. The mechanism of immortality of cancer cells and immortalized “normal” human cells is very different. Many (most?) immortalized human cells, such as the one we used - BJ-5ta – are immortalized by the manipulation of the telomerase – the cells we used are hTERT-immortalized embryonic foreskin fibroblasts. These cells are not behaving like transformed/malignant cells, and in fact are refractory to oncogenic transformation (Akagi et al. 2003).

In contrast, for cancer, we chose human breast squamous cell carcinoma cells HCC1806 - epithelial cell line isolated from the mammary gland of a female patient with acantholytic squamous cell carcinoma (ASCC), TNM Stage IIB, grade 2. They harbor many mutations associated with transformation (something BJ-5ta cells do not). We and others used these cells in cancer studies before (Wang et al., 2022).

You fully agree with you that likely, BJ-5ta cell’s response to cannabis extract is different than the response of normal human cells when the exposure is done in vivo. However, a combination of cancer and “normal” immortalized cells is acceptable and is standard when the toxicity to cancer cells is analyzed. We also effectively used BJ-5ta cells as a control for testing anti-cancer properties of cannabis extracts on HCC1806 cells (Kovalchuk et al., 2020).

So, while analyzing the mechanisms of toxicity of cannabis extracts in these normal cells would be interesting, this was beyond the scope of current work.

Wang, B., Li, D., Ilnytskyy, Y. et al. A miR-34a-guided, tRNAiMet-derived, piR_019752-like fragment (tRiMetF31) suppresses migration and angiogenesis of breast cancer cells via targeting PFKFB3. Cell Death Discov. 8, 355 (2022).

Kovalchuk, O, Li, D-P, Rodriguez-Juarez,  R, Golubov,  A, Hudson, D, Kovalchuk, I. (2020) The effect of cannabis dry flower irradiation on the level of cannabinoids, terpenes and anti-cancer properties of the extracts. Biocatalysis and Agricultural Biotechnology, 29, 101736.

Akagi T, Sasai K, Hanafusa H. Refractory nature of normal human diploid fibroblasts with respect to oncogene-mediated transformation. Proc Natl Acad Sci U S A. 2003 Nov 11;100(23):13567-72.

- It is not clear why the authors choose to stimulate cells by combination of pro-inflammatory mediators TNF-alpha and interferon gama. Are these cytokines activating a signaling pathways that are affected by cannabinoids or present terpenes?

 Our response:

Activation with TNFa/IFNg is very common in inflammation studies. It allows to activate broad range of cytokines as well as other inflammosome components. We and other used it is the past for inducing inflammation in vitro and in vivo and to study the anti-inflammatory effect of cannabis. We also in the past compared UVC, LPS and combination of TNFa/IFNg for the induction of inflammation and concluded that the latter combination is the most reliable.

Minor

- The short version of the abstract is missing the info about deep chemical analysis of 25 different varieties.

Our response:

We added this in the new version.

- the row 25 the symbol gama is missing

Added it.

- the row 79 – a typo – fiver

Corrected.

- Data in table 1 are missing values for deviation. Data from similar experiments should be averaged and expressed with values showing deviation of the data (SD or SEM).

We added SD.

- Titles for the figures “cancer#xx” or “normal#xx” are in my opinion unclear suggesting that 25 different cancer cell lines were tested. Authors should find other type of labels for graphs.

We have changed the titles to “Cancer, extract #16” and “Normal cells, extract #16”.

- A clarification of labels for bars and western blot samples in figures 4 and 5 is needed - It is not clear which samples were treated by which combinations.

We have added more information for Figure legends. We also used a black frame around images/graphs to combinate independent experiments together.

- Supplementary data – title “Supplementary Table 2. Concentration of terpenes in flowers” should be “Supplementary Table 2. Concentration of terpenes in in extract from flowers”

We agree and made this change.

 - Legends of tables explaining the associations (the supplementary tables 5- 8) should briefly explain what index of association of expressed and from which data.

We added this information.

 - Specification of flowers manufacturers should be included.

We added this information.

- In most cases the symbol gama for interferon gama is missing.

We added it everywhere.

- The use of abbreviations should be unified, and the list of abbreviations should be provided.

Round 2

Reviewer 2 Report

The authors improved the MS based on the raised comments. However, a few issues remain.

As the authors agreed, they did not use ‘normal’ cells but transformed cell. The testing of normal primary cells was, based on their own words, out of the scope of this work. So, they must clearly highlight the fact about the ‘normality’ of the cells. Both in the intro and result parts, it should be mentioned that these cells are transformed, not primary. The limitations in the interpretation of effects on transformed cells given in the previous response to the reviewers have to be mentioned in the discussion.

The tested flowers - Overall, the specification of different tested cultivars (types) of flowers is still completely vague. Is there a name or label for particular cultivar to better specify what type of material was used for the study? If an independent researcher likes to confirm the study, and choose, for example, cultivar number 5, how this can be specified to the seller? Without specification the whole study is, in my opinion, rather meaningless.

Author Response

Reviewer #2

As the authors agreed, they did not use ‘normal’ cells but transformed cell. The testing of normal primary cells was, based on their own words, out of the scope of this work. So, they must clearly highlight the fact about the ‘normality’ of the cells. Both in the intro and result parts, it should be mentioned that these cells are transformed, not primary. The limitations in the interpretation of effects on transformed cells given in the previous response to the reviewers have to be mentioned in the discussion.

 Our response:

We agree, and we mentioned this in results and discussion.

The tested flowers - Overall, the specification of different tested cultivars (types) of flowers is still completely vague. Is there a name or label for particular cultivar to better specify what type of material was used for the study? If an independent researcher likes to confirm the study, and choose, for example, cultivar number 5, how this can be specified to the seller? Without specification the whole study is, in my opinion, rather meaningless.

Our response:

We have received this information from the producers and added new Supp Figure.